# Functionalization of Polyhydroxyalkanoates (PHA)-Based Bioplastic with Phloretin for Active Food Packaging: Characterization of Its Mechanical, Antioxidant, and Antimicrobial Activities

**DOI:** 10.3390/ijms241411628

**Published:** 2023-07-19

**Authors:** Seyedeh Fatemeh Mirpoor, Giuseppe Tancredi Patanè, Iolanda Corrado, C. Valeria L. Giosafatto, Giovanna Ginestra, Antonia Nostro, Antonino Foti, Pietro G. Gucciardi, Giuseppina Mandalari, Davide Barreca, Teresa Gervasi, Cinzia Pezzella

**Affiliations:** 1Department of Chemical Sciences, University of Naples Federico II, 80126 Naples, Italy; fatemeh.mirpour.fm@gmail.com (S.F.M.); iolanda.corrado@unina.it (I.C.); giosafat@unina.it (C.V.L.G.); cpezzella@unina.it (C.P.); 2Department of Chemical, Biological, Pharmaceutical and Environmental Science, University of Messina, 98166 Messina, Italy; giuseppe.patane@studenti.unime.it (G.T.P.); giovanna.ginestra@unime.it (G.G.); antonia.nostro@unime.it (A.N.); giuseppina.mandalari@unime.it (G.M.); 3CNR IPCF, Istituto per i Processi Chimico-Fisici, Viale F. Stagno D’Alcontres 37, 98156 Messina, Italy; antonino.foti@cnr.it (A.F.); pietrogiuseppe.gucciardi@cnr.it (P.G.G.); 4Department of Biomedical and Dental Sciences and Morphofunctional Imaging, University of Messina, 98125 Messina, Italy

**Keywords:** phloretin, Listeria monocytogenes, active packaging, food contact materials, bioplastics

## Abstract

The formulation of eco-friendly biodegradable packaging has received great attention during the last decades as an alternative to traditional widespread petroleum-based food packaging. With this aim, we designed and tested the properties of polyhydroxyalkanoates (PHA)-based bioplastics functionalized with phloretin as far as antioxidant, antimicrobial, and morpho-mechanic features are concerned. Mechanical and hydrophilicity features investigations revealed a mild influence of phloretin on the novel materials as a function of the concentration utilized (5, 7.5, 10, and 20 mg) with variation in FTIR e RAMAN spectra as well as in mechanical properties. Functionalization of PHA-based polymers resulted in the acquisition of the antioxidant activity (in a dose-dependent manner) tested by DPPH, TEAC, FRAR, and chelating assays, and in a decrease in the growth of food-borne pathogens (*Listeria monocytogenes* ATCC 13932). Finally, apple samples were packed in the functionalized PHA films for 24, 48, and 72 h, observing remarkable effects on the stabilization of apple samples. The results open the possibility to utilize phloretin as a functionalizing agent for bioplastic formulation, especially in relation to food packaging.

## 1. Introduction

The formulation of eco-friendly biodegradable packaging produced from bio-based resources has received great attention during the last decades as an alternative to traditional widespread petroleum-based food packaging. In addition to their passive protective function, they can also play a pivotal role as a delivery system for bioactive compounds able to increase the mechanical characteristics of the materials and supply protection to packed foods against oxidative damage and bacterial contamination [1,2]. Among biodegradable and biobased polymers, polyhydroxyalkanoates (PHA) are one of the most investigated molecules. They are polyesters of 3-hydroxyalkanoic acids, synthesized by numerous classes of microorganisms as intracellular carbon and energy storage granules, usually under nutrient-limiting conditions [3]. Different bacterial species are known as PHA producers, displaying diversity in the type of polymer produced and the metabolic pathways that channel 3-hydroxyalkanoic acid precursors in the synthesized polymer. As a fact, PHA has been classified according to its monomer chain length into short chain length (scl)-PHA (C4 and C5), medium chain length (mcl)-PHA (C ≥ 6), and long chain length (lcl)-PHA (C > 14). Among them, polyhydroxybutyrate (PHB), a scl-PHA, is by far the most well-known PHA polymer, accumulated to up to 90% of cell dry weight by native (*Cupriavidus necator* as the workhorse) and recombinant microorganisms. On the other hand, most mcl-PHA are produced from *Pseudomonas* sp., whilst scl-mcl copolymers are produced by many bacterial species. The introduction of other monomeric units in the PHB backbone allows to improve polymer properties in favor of a reduced stiffness, higher elongation to break, and lower melting point. Although up to 13 different routes allowing channeling0specific precursors into PHA have been described, three main pathways related to sugar catabolism, fatty acids oxidation, and synthesis are responsible for the synthesis of scl-PHA, scl-mcl copolymers, and mcl-PHA, depending on the supplied C-source (sugars or lipids) [4]. Being totally produced by various bacterial species through microbial fermentation of different C-sources (both carbohydrate and lipid ones), PHA origin is properly renewable. The formulation of a plethora of PHA-based materials, i.e., copolymers, physical and/or melt-reactive blending with tailored polymers, and the addition of organic and inorganic fillers and plasticizers, has led to the improvement of mechanical features, wide processability windows, and enhanced stability and permeability performances [2]. So far, owing to its special polymer features, PHA with diverse structures and properties has been exploited in several application fields, i.e., as bioplastics for food packaging [3], in tissue engineering for biomedical implants and drug delivery carriers [5], and in the agriculture sector for the controlled delivery of herbicides [6]. 

Phloretin (dihydronaringenin or phloretol) is a dihydrochalcone, belonging to the class of flavonoids and found in many fruits and vegetables, and it is the most abundant compound identified in apples and in apple-derived products as well as in the kumquat in its glycosylated forms, characterized by the presence of the pharmacophore 2,6-dihydroxyacetophenone, which is responsible for its biological potential. Moreover, the two aromatic phenol rings A and B, the hydroxyl groups in positions 2, 4, 4′, and 6, and the carbonyl group in position 1″ supply its specific functions. Recently, the broad spectrum of beneficial properties of phloretin for human health, such as anti-inflammatory, antimicrobial, anti-hypertensive, antioxidant, anti-cancer, and other biological activities, have been reported by the authors [7,8,9]. It is also able to induce change in ion transport across lipid bilayer membranes, to regulate glucose uptake at the intestinal level exhibiting potential food–drug interactions by inhibiting human UDP-glucuronosyltransferases in vitro and anti-metabolic syndrome [10,11].

In this scenario, the aim of this work was to manufacture novel active packaging based on PHAs functionalized with phloretin able to act against food-borne pathogen bacteria and to prevent food spoilage. Amongst food-borne bacteria, *Listeria monocytogenes*, responsible for listeriosis, ranks third in the total number of deaths, exceedingly even *Salmonella* spp. and *Clostridium botulinum* (“Campylobacter and Listeria infections still rising in the EU—say EFSA and ECDC—European Food Safety Authority”, www.efsa.europa.eu, accessed on 17 December 2015). Some of its virulence factors include the ability to grow at 4 °C and reproduce inside the host’s cells. On the other hand, biodegradable packaging materials endowed with antioxidant activity are widely searched by the food industry to prevent food losses while addressing the circularity criteria [12].

Hence, in this paper phloretin-functionalized PHA materials were prepared for their potential application as food contact materials. The films were characterized in terms of functional as well as biological properties. Furthermore, their positive effect on fruit quality was investigated in apple samples. A summary of the experiment was performed and the aims obtained are depicted in Figure 1.

## 2. Results and Discussion

### 2.1. Evaluation of the Antimicrobial Activity of Phloretin

Determination of the MIC and MBC values of phloretin against several food-borne pathogens was carried out (Table 1). Gram-negative bacteria were overall more resistant compared with the positive ones. The tested dehydrochalcone was particularly effective against *L. monocytogenes* ATCC 13932 and all the *L. monocytogenes* food isolates. The activity reported for *L. monocytogenes* strains was always bacteriostatic rather than bactericidal. These data confirmed our previous results [7] and led us to test phloretin for the subsequent film functionalization.

We have previously demonstrated the efficacy of polyphenols against *L. monocytogenes* strains: A bactericidal effect against a food isolate was detected using an extract of *Hibiscus sabdariffa* L., rich in anthocyanins (cyanidin-3-O-sambubioside and delphinidin-3-O-sambubioside), whereas a bacteriostatic and bactericidal effect was obtained with polyphenols-rich extracts from nuts [13,14,15]. Due to the dangerous effects of *L. monocytogenes* infections, which can cause listeriosis often with severe outbreaks, and the possibility of contamination during food chain production, packing, and distribution, the obtained results show phloretin as a promising functionalizing agent for the production of bioplastic in food packaging industrial applications, and this may help control contamination in the food chain, from production to end consumers.

### 2.2. Preparation and Characterization of Phloretin-Functionalized PHA Films

The films were produced by casting 200 mg of PHA polymer blend (90:10 PHB/mcl-PHA) in the absence and in the presence of different concentrations of phloretin (5–7.5–10–20 mg/film). They appeared macroscopically very similar, whitish, and very homogeneous. Their functional and biological characterization is described in the following paragraphs. 

#### 2.2.1. Mechanical Properties

The mechanical properties reported in Figure 1 indicate that the tensile strength (TS) of PHA films significantly increased when the films contain 5 and 7.5 mg of phloretin compared to the neat PHA film. However, TS progressively decreased in the presence of a higher amount of this phenolic molecule (10 and 20 mg). The film’s Young’s modulus (YM) showed a similar behavior, increasing after phloretin grafting in the film with the lower concentration and decreasing at higher amounts of phloretin. However, it should be mentioned that the elongation at break (EAB) of all the films is lower than 2% and the thickness was not affected by different concentrations of phloretin remaining similar for all the films. Similar mechanical behaviors were reported by Figueroa–Lopez et al. [16] and Rubini et al. [17], who demonstrated that the presence of low contents of eugenol and quercetin incorporated in poly(3-hydroxybutyrate-co-3-hydroxyvalerate) and gelatin films were able to improve all the mechanical properties following by worsening this property at higher concentrations. This behavior could be mostly related to the not homogeneous distribution of the additives in the film matrix that should be overcome by means of dispersant agents (e.g., surfactants and plasticizers produced from epoxidized oils) [18]. 

#### 2.2.2. Water Sensitivity and Opacity

The results reported in Table 2 indicate that moisture content and swelling ratio values of the films decreased as a function of phloretin at a lower concentration, which was significantly lower than the control film and slightly lower than the films containing the higher amount of phloretin. The reduction in moisture content and swelling ratio at lower concentrations of phloretin most likely is due to the hydrophobic properties of this phenol that limited the water retention in the film matrix [19]. However, all the films have low moisture content (less than 6%) and swelling ratio (less than 4%). Although there is not a clear trend as a function of phloretin amount, the addition of molecule to PHA-based films determines an increase in film hydrophobicity as depicted by the increase in contact angle values. This effect can be attributed to the hydrophobic nature of phloretin which positively contributes to the change wettability of the surface. Another important film property is film opacity, as it is one of the key factors that affect the food quality of a packed food and consumers’ willingness to choose a certain type of product [20]. From Table 2 it is clear that the films become opaquer as a function of enhancement of the phloretin content.

#### 2.2.3. FTIR Characterization

The ATR-FTIR spectra obtained for phloretin, PHA-based film, and PHA/phloretin film (the film loaded with 7.5 mg of phloretin was chosen as an example) are reported in Figure 2. In particular, the spectra of PHA and PHA/phloretin films show peaks between 2973–2962 cm^−1^ that can be assigned to the stretching vibration due to asymmetric CH_2_ of the lateral monomeric chains and to the symmetrical methyl group of the polymer matrix. The sharp intense band at 1716 cm^−1^ is typical of ester carbonyl groups (C=O) of polymers. However, a broad but less intense shoulder peak at about 1744 is detectable. This can be attributed to the greater vibration energy of the amorphous ester carbonyl group domain. Moreover, in the region 1500–1650 cm^−1^ of the PHA/phloretin sample, the characteristic stretching vibration of the ester carbonyl group of phloretin is clearly detectable. However, the spectrum revealed no real interaction among components but rather the dispersion of active compounds into the polymer matrix (Figure 2).

#### 2.2.4. Raman Spectroscopy of PHA and PHA/Phloretin Film

Figure 3a compares the Raman response of PHA and PHA/phloretin films, which put in evidence a slight increase in fluorescence background for both excitation wavelengths, even if the fluorescence intensities are much higher at 638 nm (Figure 3b). Moreover, it is worth noting that PHA film shows a significant fluorescence emission even without phloretin when the excitation wavelength is set at 638 nm (black line in Figure 3b). However, the shape of the fluorescence emission is different when phloretin is also inside the film, pointing out its contribution to the total emission (red line in Figure 3b) even when the excitation line is matching only the tail of the absorption band [21].

In order to highlight the vibrational modes of the PHA/phloretin with respect to pure PHA, the fluorescence background was subtracted from the spectra and they were normalized to the vibrational mode of PHA at 1723 cm^−1^ (Figure 4a). Figure 4b is showing the vibrational Raman modes of phloretin obtained by subtracting the Raman contributions of PHA (black line in Figure 3a) from the Raman spectrum of PHA/phloretin (red line in Figure 4b), which is compatible with what reported from other studies [21,22]. In the high frequency region, it is clearly visible the aromatic ring (C–H) stretching vibration of phloretin at around 2930 and 3060 cm^−1^ or C–O stretching bands at around 1622 cm^−1^ [23]. On the other side, the bands at 737 and 1200 cm^−1^ are ascribed to the O-C-C and C-C-C bending modes, respectively [23]. Finally, in the range between 250 and 650 cm^−1^ we can detect other Raman peaks due to the torsional modes of phloretin [23].

### 2.3. Release Assay

The analysis of the PHA/phloretin films upon incubation for different times with methanol, revealed the presence of the bioactive molecule released from the bioplastic. In fact, reverse-phase high-performance liquid chromatography with diode-array detection (RP-HPLC-DAD) analysis of the phloretin released from the material, detected at 278 nm, revealed the presence of a well-defined peak corresponding to the dihydrochalcone (Figure 5). The identification of the compound was also confirmed by the analysis of the same separation performed with the authentical standard and by spiking the starting material with the same compound. The results show that the release is independent of the concentration of phloretin present in the functionalized bioplastic and followed the same pattern. After approximately 24 h, the 80 ± 2% of phloretin was released, and in the following 24 h, a further 4–8% of the molecule was found in methanol. In the next 48, 72, and 96 h, the release did not change. The mechanical fragmentation led to a further release of phloretin of about 5–7%, reaching a total amount of phloretin incorporated in the film ranging between 92–95% of the starting concentration utilized for the preparation of the functionalized bioplastic. Using phosphate saline buffer (PBS), as release medium, no release was found.

Using PBS as a release medium, no release was reported.

### 2.4. Antioxidant Assay

In vitro biological assays (DPPH, TEAC, FRAP, ferrozine assay) were performed to evaluate the antioxidant activity of PHA films functionalized with different amounts of phloretin. Reactive oxygen species are dangerous molecules that can drastically change the quality of the food and its half-life, beyond being dangerous for living organisms that consume these matrices. Almost all tests showed that the free radical scavenging activity of PHA films was due exclusively to the presence of phloretin. The bioactive role of phloretin was first highlighted by using the DPPH assay (Figure 6A). It was observed that the free radical scavenging activity of PHA film was due exclusively to the presence of phloretin, considering that the film not functionalized did not show any activity. The reduction in the radical levels was found to be in a dose-dependent manner, reaching the maximum activity in the samples prepared with 10 mg of phloretin. Using the TEAC assay, it was observed that the films not functionalized did not show any activity (Figure 6B); instead, the films functionalized with 5 mg promoted about 80% of residual absorbance reduction and it increased to 90% when the amount of phloretin used for the functionalization was higher (7.5–10 and 20 mg). The same assumption was obtained with the FRAP assay, given that the ability of the films to reduce the ferric ion (Fe^3+^)–ligand complex to the intensely blue-colored ferrous (Fe^2+^) complex was observed only with the PHA film functionalized with 7.5, 10, and 20 mg of phloretin and a certain dose-dependent activity was observed increasing the amount of phloretin. While no activity is reported for the PHA film without functionalization or with an amount of flavonoid of 5 mg (Figure 6C). The ferrozine assay showed the chelating capacity of all the PHA films, and this activity increased in a dose-dependent manner with the presence of the flavonoid (Figure 6D). The different antioxidant tests let us shed some light on the antioxidant potentials of the functionalized films and, in particular, for the presence of phloretin. This latter is a natural antioxidant derived from plant sources and it plays an important role by directly scavenging free radicals or increasing antioxidant defenses and decreasing, in general, the process of oxidation due to the presence of reactive oxygen species (ROS). The film functionalized with phloretin is able to perform antioxidant activity in both types of categories based on the chemical reaction for the scavenging of ROS: electron transfer (ET) reaction-based assays and hydrogen atom transfer (HAT) reaction-based assays. Thus, the functionalized films are able to scavenge radicals that require both the electron or hydrogen atom donating capacity due to the versatility of phloretin and of its pharmacophore (2,6-dihydroxyacetophenone). This is of particular interest because different kinds of ROS can be produced during food process transformation, conservation, and commercialization, decreasing its quality and shelf-life.

### 2.5. Evaluation of Phloretin-PHA Antimicrobial Activity

The diffusion assay showed no inhibition zone for PHA films at all concentrations used. However, no growth was observed under all the films regardless of the presence of phloretin. The lack of inhibition halo was in accordance with the release test, which did not report phloretin release in the water solvent and explains the reason for the lack of antimicrobial properties around the film. The PHA films have been tested by incubation in 1 mL of the microbial liquid culture at the concentration of 5 × 10^5^ CFU mL^−1^. In line with our previous results on phloretin [7], the best activity was observed against *L. monocytogenes*, with a reduction of about 3-log compared with the control. Moreover, a dose-dependent effect was observed (Figure 7). It has been found that the antibacterial activity of phloretin may be attributable mainly to the ability to increase cell membrane permeability and aggregate nuclear acid materials [24]. No activity was reported for *E. coli*, but antagonism activity was observed using the films functionalized with 10 and 20 mg of phloretin, showing an increase in the microorganism growth. A negligible antimicrobial activity was, instead, detected for *S. enterica* serovar Typhimurium ATCC 13311. These results should be analyzed, taking into account that the phloretin was not released in the culture medium and the antimicrobial activity was limited to the microorganisms near the films. Moreover, the lack of activity against *E. coli* correlates with the MIC determination (Table 1), where no effect was detected at the maximum tested concentration. On the other hand, the high MIC and MBC values obtained with *S. enterica* serovar Typhimurium ATCC 13311 could explain the lack of activity of the functionalized films against these strains, given the phloretin concentration in the films did not reach the MIC values. The severity of *L. monocytogenes* disease indicates that safe handling of food is of paramount significance to ensure public health. Analyzing these data, it is important to mention that listeriosis was the fifth most reported zoonosis in the EU in 2021, with an increase in notification rate of 14% compared with 2020 in the EU as reported in “The European Union One Health 2021 Zoonoses Report” edited by EFSA and ECDC (available at https://efsa.onlinelibrary.wiley.com/doi/epdf/10.2903/j.efsa.2022.7666, accessed on 14 July 2023). *Listeria monocytogenes* contamination can be found in different foods (such as raw vegetables, cheeses, especially soft cheeses, meats, and smoked fish). The functionalized films can decrease the growth of *L. monocytogenes*, especially at the maximum tested concentrations, creating an interesting new active food packaging bioplastic. Taking into account that cooking at high temperatures kills the bacteria, but contamination of foods can happen also after production before it is packaged, the creation of active food packaging materials able to reduce the growth of *L. monocytogenes* can be an important step to reduce its proliferation. Moreover, even with the forming biofilm ability, new strategies to control the growth of *Listeria monocytogenes* are warranted. Regarding the biofilm formation, the phloretin-functionalized PHA films showed an inhibitory effect against *L. monocytogenes*. In particular, a biomass reduction compared to neat films of about 13%, 20%, and 33% was found with PHA films containing 7.5, 10, and 20 mg of phloretin, respectively. A negligible activity was instead detected against *E. coli* and *S. enterica* serovar Typhimurium ATCC 13311.

### 2.6. Food Fresh-Keeping Test

In order to study their fresh-keeping performance, the prepared films were used to pack fresh apple samples, as shown in Figure 8. These representative images of the apple samples were digitalized with a photo-camera and elaborated with the program Image J in function of their quality change due to browning reaction after different days of incubation (Figure 8a). It can be seen that apple samples exposed to air showed a browning phenomenon evident after 24 h, but marked after 48 and 72 h. The apple samples packed with a PHA bag functionalized with 20 mg of phloretin showed only limited color change after 24 and 48 h, but an obvious slight browning phenomenon can be observed after 72 h. In the presence of neat PHA, the samples packed show browning effects similar or superior to the samples not packed. To have a more accurate measure of the browning reaction we have monitored the changes in absorbance at 420 nm. As can be seen in the graph (Figure 8b) the changes in absorbance are almost completely superimposable with the data obtained by computer elaboration of the images. The changes in quality of the apple samples were also analyzed as a function of weight loss, change in antioxidant activity, and sugar content as a function of brix degrees. Figure 8c shows the weight loss of the apple samples after 24, 48, and 72 h. As can be seen, the PHA pack functionalized with 20 mg of phloretin showed a lower weight loss than those without a pack or with only a PHA bag for up to 48 h. After 72 h, the weight loss was the same for all the tested samples. A similar trend can be seen also in the monitoring of brix degrees changes with the samples packed in the functionalized PHA film showing the best performance (Figure 8d). The influences of the presence of phloretin in the functionalization of the PHA film are evident in the monitoring of antioxidant activity with the DPPH assay. As can be seen in the graph (Figure 8e), the apple samples packed in the PHA functionalized with phloretin maintained almost the same antioxidant activity also after 72 h of incubation, while in the other two samples, there is a decrease in this potentiality that is most marked in the samples not packed, that after 72 h lost about the half of its activity. The apple samples packed only with PHA film showed a decrease in this activity but are inferior to the ones not bagged and clearly superior to the ones fortified with phloretin. The analysis of the obtained results in the different tests shows that PHA-phloretin functionalized films may have food-preservation performance superior to the only film in agreement with the data characterization data obtained in the above section. In particular, the data obtained in the antioxidant determinations are in line with the one obtained in this section and are correlated with the ability of phloretin to scavenge radicals based on ET and HAT mechanism, preserving the endogenous antioxidant of the food and decreasing the browning reactions. 

## 3. Materials and Methods

### 3.1. Reagents and Standard Solutions

HPLC-grade acetonitrile and methanol were supplied by Sigma-Aldrich (St. Louis, MO, USA), dimethylformamide (DMF) by Carlo Erba (Milano, Italy). Phloretin (≥99%) was supplied by Sigma-Aldrich (St. Louis, MO, USA) and was used as standard. Mueller–Hinton broth and agar were supplied by Oxoid (Sigma, Milano, Italy).

### 3.2. Production and Characterization of Polyhydroxyalkanoates Based-Films

#### 3.2.1. Polymer Production

For the production of poly(3-hydroxybutyrate) (PHB), microbial fermentation of *Cupriavidus necator* DSM 428 was carried out in a 5L BioFlo/CelliGen^®^115 (Eppendorf New Brunswick) following a two-step growth protocol as described in Mirpoor et al. [6]. Briefly, to induce polymer accumulation, the first step of growth in rich medium (TSB, Tryptic Soy Broth) for 24 h at 30 °C was followed by an additional 24 h in minimal medium MMCn (Budde, 2011) containing 20 g/L fructose. Fermentation parameters were set as follows: inoculum at 0.1 OD600, agitation rate 220 rpm, and the DO concentration maintained at 30% of air saturation. After 72 h the cells were harvested by centrifugation (6000 rpm, 20 min), lyophilized, and the polymer extracted according to Turco et al. [25]. Medium chain length PHA (mcl-PHA) was produced by microbial fermentation of *Pseudomonas resinovorans* NRL B-2649 in minimal medium E supplemented with 0.6% *v*/*v* oleic acid as C-source [6]. Fermentation parameters were set as follows: inoculum at 0.1 OD600, agitation rate 220 rpm, and the DO concentration maintained at 30% of air saturation. The cells were harvested after 48 h, and the polymer was extracted according to Turco et al. [25].

#### 3.2.2. Film Preparation

PHA-based films were obtained by the solvent-casting method. PHA solutions were prepared by dissolving the polymers in chloroform at a concentration of 20 mg/mL. The binary blend was formulated at a 90:10 PHB/mcl:PHA ratio, for a total amount of 200 mg of polymers in a volume of 20 mL. Phloretin was dissolved in acetone and added to the polymer solutions to rich a final amount of 5, 7.5, 10, and 20 mg/film. The solutions were cast into glass Petri dishes (diameter of 10 cm), which were kept at room temperature until complete solvent evaporation occurred. Slow solvent evaporation was performed in a saturated chloroform atmosphere to avoid the formation of cracks and non-selective voids in the films and to guarantee their homogeneity.

#### 3.2.3. Characterization of PHA-Based Films

Mechanical properties

Mechanical properties of the films were determined by measuring tensile strength (TS), Young’s modulus (YM), and elongation at break (EAB) according to ASTM standard method D882-12 (Testing & Materials–ASTM, 2012) using an Instron universal testing instrument (Model 5543 A, Instron Engineering Corp., Norwood, MA, USA). The film strips were cut into the dimension of 1 × 8 cm^2^ and placed between the grips of the machine. The initial grip separation and cross-head speed were set to 40 mm and 5 mm/min, respectively.

Moisture content, swelling ratio, and contact angle

Moisture content, swelling ratio, and contact angle of the films were measured according to the method described by Mirpoor et al. [26] with slight modification. The moisture content of the films (3 × 3 cm^2^) was calculated by weighing the films before and after oven drying at 105 °C and dividing the weight loss due to oven drying by the film’s initial weight. In order to measure the film swelling ratio, the initial weight (*Wi*) of the film samples (3 × 3 cm^2^) was recorded and then immersed in 30 mL distilled water at 25 °C for 1 h. After that, films were removed and weighed after drying the surface water of the films with absorbent paper (*Ws*). The swelling ratio was calculated as follows:(1)Swelling ratio (%)=(Ws−Wi)×100/Wi

The contact angle was measured by using a homemade contact angle goniometer. The film strip was placed on the horizontal stage and then 10 µL of distilled water was deposited on both sides of each film at different points. The image of the water drop was captured at the moment that the drop was in contact with the film surface. The mean value of the contact angle was measured with ImageJ software. Five measurements were reported as the average contact angle value.

Opacity

To measure the film opacity, the absorbance of the four strips of each film (1 × 4 cm) was measured at a wavelength of 600 nm by using a UV–Vis spectrophotometer (SmartSpec 3000 Bio-Rad, Segrate, Milan, Italy) according to the method of Jahed, Khaledabad, Bari, and Almasi [27]. Then, the film opacity was calculated by dividing the obtained absorbance value by the film thickness (mm).

Fourier transform infrared spectroscopy (FTIR-ATR)

FTIR analysis was performed in ATR (attenuated total reflection) modality. All the samples were analyzed at room temperature with an FTIR Nicolet 5700 spectrometer (Thermo Fisher Scientific, Waltham, MA, USA). Spectra were recorded as an average in a range of 4000–600 cm^−1^, with a spectral resolution of 2 cm^−1^. Spectra analysis was done with OMNIC 9 software (Thermo Fisher Scientific, Inc., Waltham, MA, USA).

Raman spectroscopy of PHA and PHA/Phloretin film

Raman spectroscopy was carried out with an Xplora Plus microspectrometer (Horiba Scientific, Singapore) equipped with 638 and 785 nm diode lasers. The incident light was focused onto the sample surface using a 100× objective (laser power was set at 38 and 10.2 mW for 785 and 638 nm, respectively). The Raman signal was collected in a backscattering configuration through the same objective and dispersed by a 1200 L/mm diffraction grating onto a CCD detector (Sincerity–Horiba Scientific) using an integration time of 10 s.

### 3.3. Antimicrobial Assays

#### 3.3.1. Microbial Strains and Culture Conditions

The Gram-positive bacteria used were *Enterococcus hirae* ATCC 10541, *Listeria monocytogenes* ATCC 13932, and 14 food isolates of *L. monocytogenes* belonging to serotypes 1/2a (4 strains), 1/2b (4 strains), 1/2c (3 strains) and 4b (3 strains). The Gram-negative bacteria used were *Escherichia coli* ATCC 10536, *Salmonella enterica* serovar Typhimurium ATCC 13311, and two clinical isolates of *S. enterica* serovar Typhimurium and *S. enterica*. All bacteria were grown in Mueller–Hinton broth (MHB, Oxoid, CM0405, Sigma, Italy) except for *L. monocytogenes* strains, which were grown on tTyptic Soy broth (TSB; CM0129, Oxoid, Sigma, Italy) at 37 °C (18–20 h). 

#### 3.3.2. Susceptibility Studies of Phloretin 

The minimum inhibitory concentration (MIC) and the minimum bactericidal concentration (MBC) of phloretin were determined by the broth microdilution method according to CLSI for bacteria [28]. The MIC value was considered as the lowest concentration of phloretin giving a complete inhibition of visible bacterial growth after incubation for 24 h. The MBC value was defined as the lowest concentration of phloretin able to kill 99.9% of the inoculum after 24 h. The antimicrobial activity of PHA films was evaluated using both an adapted diffusion test and a broth microdilution assay for bacteria that grow aerobically, as recommended by the CLSI [28,29]. 

#### 3.3.3. Disc Diffusion Assay of PHA Films

The PHA films were sterilized under a UV lamp before each assay. The diffusion test was performed as recommended by the CLSI 2018 [29]. Briefly, each film (1 cm^2^), functionalized with 5, 7.5, 10, and 20 mg of phloretin, was deposed on plates of Mueller–Hinton Agar (MHA, Oxoid, CM0405, Sigma, Italy) or tryptic soy agar (TSA; CM0131, Oxoid, Sigma, Italy) previously swabbed with the standardized inoculum (5 × 10^8^ CFU/mL). The susceptibility of bacteria to the functionalized films was measured as the clear area that appeared around and/or under the piece of film. Neat PHA films were included as controls. 

#### 3.3.4. Antibacterial Activity of PHA Films and Biofilm Biomass Measurement 

Each PHA film (1 cm^2^) functionalized with 5, 7.5, 10, and 20 mg of phloretin was incubated in a medium (1 mL) containing a standardized bacterial load at the concentration of 5 × 10^5^ CFU/mL. The bacterial growth was evaluated after 24 h at 37 °C by plating serial dilutions. Plates were incubated at 37 °C for 24–48 h, and the number of colonies was reported as CFU mL^−1^. Neat PHA films were included as controls. In addition, for the investigation of anti-biofilm properties, after 24 h-incubation the biofilm formed by the tested bacteria on the PHA films was evaluated by biomass measurement. The films were washed with PBS, dried, and stained with 0.1% safranin, as previously reported [30]. The stained biofilms were suspended in 30% acetic acid aqueous solution and the mean optical density at 492 nm (OD_492_) was measured. The reduction percentage of biofilm was calculated using the following equation: (2)$$Biofilm\;Reduction\;(\%)=100-({OD}_{492}\;PHA\;with\;phloretin)/({OD}_{492}\;neat\;PHA)\times 100$$

### 3.4. Identification of Phloretin Release 

Prior to analysis, 1 mL of solvent (methanol or PBS) was added to each PHA film (1 cm^2^) and incubated at room temperature. The solvent was then filtered through an Iso-Disc P-34, 3 mm diameter PTFE membrane, 0.45 μm pore size (Supelco, Bellefonte, PA, USA). The evaluation of the active molecule released from PHA films was performed by reverse-phase HPLC-DAD detection, using a Shimadzu HPLC system equipped with a UV–Vis photodiode array detector (DAD; Shimadzu, Kyoto, Japan) and a fluorescence detector (Hewlett Packard) according to Barreca et al. [31] at different intervals of time. Finally, the samples were mechanically homogenated with a mortal in the solvent utilized for release to obtain the maximum amount of phloretin present in the films. The column was a Discovery C18 (250 × 4.6 mm i.d., 5 μm) supplied by Supelco (Bellefonte, PA, USA), equipped with a 20 × 4.0 mm guard column, and held in a column oven set at 30 °C. The injection loop was 20 μL, and the flow rate was 1.0 mL/min. The mobile phase consisted of a linear gradient of acetonitrile/H_2_O as follows: 5–20% (0–15 min), 20–30% (15–20 min), 30–100% (20–35 min), 100% (35–40 min), 100–5% (40–45 min), and 5% (45–55 min). UV spectra were recorded between λ 200 and 450 nm, and simultaneous detection by diode array was performed at λ 278 nm. Nitrogen was used as a sheath gas with a flow of 50 arbitrary units. Peak identity was confirmed by comparing the retention time and absorption spectra with the one of pure (≥99%) commercially available standard (concentration range 1–50 ug/mL). Each sample was tested three times and gave superimposable chromatograms.

### 3.5. Antioxidant Assays

2,2-Diphenyl-1-picrylhdrazyl (DPPH) and 2,2′-azino-bis (3-ethylbenzothiazoline-6-sulfonic acid) (ABTS) radical scavenging assays, ferric reducing power (FRAP), and the ferrozine assay were used to investigate the in vitro antioxidant efficacy of PHA films.

#### 3.5.1. DPPH Assay

The free radical method was performed according to Barreca et al. [31], in short, by using the stable free radical 2,2-diphenyl-1-picrylhydrazyl (DPPH^•^). Briefly, each 1 cm^2^ PHA film was mixed with 80 μM DPPH^•^ in methanol in a final volume of 1 mL. The variations in absorbance at 517 nm were monitored over 30 min with a Varian Cary 50 UV–Vis spectrophotometer. The concentration of DPPH radical in the cuvette (1.0 cm path length) was chosen to give absorbance values less than 1.0. The inhibition (%) of radical scavenging activity was calculated by the following equation:(3)I (%)=[(Ac−As)/Ac]×100
where *Ac* is the absorbance of the control and *As* is the absorbance of the sample. All tests were run in triplicate and the results were expressed as means ± standard deviation (SD). 

#### 3.5.2. ABTS Radical Scavenging Assay

The 2,2′-azino-bis(3-ethylbenzothiazoline-6-sulphonic acid ABTS free radical-scavenging activity was carried out by a decolorization assay according to Smeriglio et al. [32]. The radical cation ABTS^•+^ was added to each 1 cm^2^ film, and the absorbance changes at 734 nm were recorded in a spectrophotometer after 6 min. 

#### 3.5.3. Ferric Reducing Antioxidant Power (FRAP)

The FRAP assay was performed according to Barreca et al. [31]. The fresh working FRAP reagent was prepared daily by mixing 25 mL of acetate buffer (300 mM, pH 3.6), 2.5 mL of 2,4,6-Tris(2-pyridyl)-S-triazina (TPTZ) solution (10 in 40 mM HCl), and 2.5 mL of FeCl_3_ (20 mM). The reagent was warmed up to 37 °C; and then 1500 μL were placed in a cuvette (1.0 cm path length) and the initial absorbance was read. Each 1 cm^2^ PHA film was added to the cuvette and the absorbance was measured after 4 min at wavelength 593 nm with a Varian Cary 50 UV–Vis spectrophotometer. All tests were run in triplicate and the results were expressed as means ± standard deviation (SD). 

#### 3.5.4. Ferrozine Assay

The potential chelating activity of 1 cm^2^ PHA films toward ferrous ions was analyzed according to Papalia et al. [33] with little modifications. EDTA (0.1 mM final concentration) was used as a reference compound. PHA films were added to a solution of 0.5 mM FeSO_4_ (0.01 mL). After the addition of 5.0 mM ferrozine (0.4 mL) solution, the samples were shaken and left for 10 min at room temperature (RT). Finally, the absorbance at 562 nm of the solution was measured with a spectrophotometer. The inhibition (%) of ferrozine Fe^2+^ complex formation was found using the following equation:(4)% Inhibition=[(Ac−As)/Ac]×100
where *Ac* is the absorbance of the control and *As* is the absorbance of the samples where the phloretin should be released.

### 3.6. Analysis of Food Preservatives Properties

#### 3.6.1. Apple Samples Design

We selected apples (*Malus domestica*, variety Pink Lady) that were consistent in size (medium fruit), color, and maturity and free from pests, diseases, and mechanical damage purchased from the local market. At room temperature, the apples were denied of the peel and cut in to pieces of 1.2 × 1.2 × 0.6 (L × W × H) cm with a professional cutter stainless steel with 1/2- and 1/4-inch blades, obtaining a weight of 1.5 ± 0.12 g for each apple sample. Each operation was performed under a biological fume hood to work under sterile conditions and all the following procedures were performed in the same conditions. Each sample of apple was packaged in a single-layer film, and the film was sealed and closed manually. In this study, two groups of self-made biodegradable plastic films were used to pack the apple samples, and filmless packaging was used as the control. The tests were carried out for several consecutive hours, maintaining the samples at a controlled room temperature of 25 °C, and every 0, 24, 48, and 72 h, three samples from each group were withdrawn and utilized for the following analysis.

#### 3.6.2. Computer and Graphic Elaboration

The changes in the appearance of the apple samples were obtaining digitalizing the images of the samples, after the different intervals of time, by a professional digital camera. The digital images have been analyzed by ImageJ software (available at https://imagej.nih.gov/ij/download.html, accessed on 10 June 2023).

#### 3.6.3. Browning Reaction and Determination of Brix Degree

The browning reaction was determined by monitoring the absorbance changes at 420 nm with UV–Vis spectrophotometry according to Xu et al. [34]. Each sample was diluted with distilled water in the ratio 1:1 (*w:v*), homogenated with a mortal pre-chilled, and centrifuged at 10,000 rpm, 4 °C for 10 min. The supernatant of each sample was utilized to monitor the changes at 420 nm. Distilled water was used as a blank control. The same samples were also utilized to analyze brix degree with a brix refractometer and changes in the antioxidant activity with the DPPH assay. For the DPPH assay, 10 μL of the samples has been withdrawn and tested as described in Section 3.5.1.

#### 3.6.4. Weight Loss

The weight loss rates were calculated by the following equation:(5)$$Weight\;loss\left(\%\right)=\frac{m_{o}-m}{m_{o}}\times 100$$
where *m_o_* and *m* are the initial weight and weight after different intervals of time (0, 24, 48, 96 h) of apple samples [35]. The experiments were triply repeated to get the average value.

## 4. Conclusions

In this paper, the manufacture of films made of PHA as renewable sources for the production of novel bioplastics was reported. Such materials for the first time were further functionalized with different amounts of phloretin, a dihydrochalcone, belonging to the class of flavonoids found in different fruits and vegetables endowed with interesting biological properties, that significantly increased the antioxidant potential and antimicrobial properties of the produced films without drastically changing mechanical and hydrophilicity features. As a fact, the phloretin-grafted materials were also able to preserve apple samples’ freshness envisaging their potential application as bio-based packaging systems to be applied for the extension of food shelf-life.

## Data Availability

Not applicable.

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
