# Peer review of "Functionalization of Polyhydroxyalkanoates (PHA)-Based Bioplastic with Phloretin for Active Food Packaging: Characterization of Its Mechanical, Antioxidant, and Antimicrobial Activities"

_ijms, 2023, doi:10.3390/ijms241411628_

Round 1

Reviewer 1 Report

ijms-2509685

Title - Functionalization of Polyhydroxyalkanoates (PHA)-based bioplastic with phloretin for active food packaging: characterization of its mechanical, antioxidant and antimicrobial activities.

The manuscript by Mirpoor et al. demonstrated strategy for PHA functionalization for potential active food packaging application. Overall, the manuscript is interesting and requires major revision before its publication in “IJMS” as follows:

Comments:

1.      Lines 36-42, please add a few citations to justify the statements.

2.      Lines 51-53, this information can be more elaborated with also examples of broad biotechnological applications of PHA such as biocontrol agents, tissue engineering, and agriculture sector. Also add minor information’s – i) PHA producers, ii) synthesis pathway, iii) suitability of co-polymers over monomers of PHA, and iv) in addition, bacteria can accumulate PHA up to 90% of their dry cell mass.

3. In Line 86, the name of the organism should be in italics, please also correct at other places in the text.

4.      Fig. 3, please labeled some important peaks in FTIR spectra.

5.      Discussions are weak, it requires significant progresses to justify the implication of this study’s experimental findings.  

6.      Please add one illustration to the summary of the present study such as functionalization of PHA, alteration in properties, mechanism of antimicrobial activity, etc.

7.      Figures quality should be improved, for example, font size, width size, and demonstration for consistent presentation. 

Minor editing of English language is required

Author Response

Title - Functionalization of Polyhydroxyalkanoates (PHA)-based bioplastic with phloretin for active food packaging: characterization of its mechanical, antioxidant and antimicrobial activities.

The manuscript by Mirpoor et al. demonstrated strategy for PHA functionalization for potential active food packaging application. Overall, the manuscript is interesting and requires major revision before its publication in “IJMS” as follows:

Comments:

  1. Lines 36-42, please add a few citations to justify the statements.

According to reviewer suggestion we have added citations.

  1. Lines 51-53, this information can be more elaborated with also examples of broad biotechnological applications of PHA such as biocontrol agents, tissue engineering, and agriculture sector. Also add minor information’s – i) PHA producers, ii) synthesis pathway, iii) suitability of co-polymers over monomers of PHA, and iv) in addition, bacteria can accumulate PHA up to 90% of their dry cell mass.

According to reviewer suggestion we have elaborated this part of the manuscript.

  1. In Line 86, the name of the organism should be in italics, please also correct at other places in the text.

Done

  1. Fig. 3, please labeled some important peaks in FTIR spectra.

Done

  1. Discussions are weak, it requires significant progresses to justify the implication of this study’s experimental findings.  

According to reviewer suggestion we have increased the discussions.

  1. Please add one illustration to the summary of the present study such as functionalization of PHA, alteration in properties, mechanism of antimicrobial activity, etc.

Done

  1. Figures quality should be improved, for example, font size, width size, and demonstration for consistent presentation.

According to reviewer suggestion we have improved the quality of the figures.

Reviewer 2 Report

This study focused on the development and testing of eco-friendly biodegradable packaging using PHA-based bioplastics functionalized with phloretin. The investigation revealed that the addition of phloretin had a mild influence on the mechanical and hydrophilicity properties of the materials. Moreover, the functionalization of PHA-based polymers with phloretin demonstrated antioxidant activity and inhibited the growth of foodborne pathogens. These findings suggest that phloretin could be a promising agent for enhancing bioplastic formulations, particularly for food packaging applications.

The only necessary correction is the formatting of references. Please carefully review the requirements and ensure consistent formatting for each reference.

In future research, it would be beneficial for the authors to expand their investigations by conducting additional tests and analyses on the long-term stability and degradation characteristics of the phloretin-functionalized PHA films. This would allow for a better understanding of the durability and shelf life of the packaging materials, particularly in different environmental settings.

Author Response

This study focused on the development and testing of eco-friendly biodegradable packaging using PHA-based bioplastics functionalized with phloretin. The investigation revealed that the addition of phloretin had a mild influence on the mechanical and hydrophilicity properties of the materials. Moreover, the functionalization of PHA-based polymers with phloretin demonstrated antioxidant activity and inhibited the growth of foodborne pathogens. These findings suggest that phloretin could be a promising agent for enhancing bioplastic formulations, particularly for food packaging applications.

The only necessary correction is the formatting of references. Please carefully review the requirements and ensure consistent formatting for each reference.

According to reviewer suggestion we have corrected the formattation of the references.

In future research, it would be beneficial for the authors to expand their investigations by conducting additional tests and analyses on the long-term stability and degradation characteristics of the phloretin-functionalized PHA films. This would allow for a better understanding of the durability and shelf life of the packaging materials, particularly in different environmental settings.

We thanks the reviewers for the useful suggestions and we will take into accounts in our future works.

Round 2

Reviewer 1 Report

Accept